# Dietary Intake of Micronutrients and Use of Vitamin and/or Mineral Supplements: Brazilian National Food Survey

**DOI:** 10.3390/nu16223815

**Published:** 2024-11-07

**Authors:** Caroline da Rosa Pavlak, Michele Drehmer, Sotero Serrate Mengue

**Affiliations:** 1Postgraduate Program in Epidemiology, School of Medicine, Universidade Federal do Rio Grande do Sul, Porto Alegre 90035-190, Rio Grande do Sul, Brazil; caroliner2007@gmail.com (C.d.R.P.); sotero@ufrgs.br (S.S.M.); 2Postgraduate Program in Food, Nutrition and Health, School of Medicine, Universidade Federal do Rio Grande do Sul, Porto Alegre 90035-190, Rio Grande do Sul, Brazil

**Keywords:** vitamin and/or mineral supplements, estimated average requirements, Brazilian National Food Survey

## Abstract

Background/Objectives: Vitamin and/or mineral supplements are designed to correct micronutrient deficiencies or maintain adequate intake. However, evidence suggests the indiscriminate use of these products, particularly among populations that already meet their micronutrient requirements through diet. This study aims to estimate the prevalence of vitamin and/or mineral supplement use and assess the dietary intake of micronutrients among users and non-users in the Brazilian adult and elderly populations. Methods: The prevalence of vitamin and/or mineral supplement use was estimated from a sample of 37,364 individuals who participated in the Brazilian National Food Survey, a module of the 2017–2018 Household Budget Survey. The average dietary intake of micronutrients—including calcium, magnesium, phosphorus, iron, copper, zinc, vitamin A, thiamine, riboflavin, niacin, cobalamin, pyridoxine, vitamin D, vitamin E, vitamin C, and folate—was calculated for both users and non-users of these supplements, based on 24 h dietary recalls collected during the survey. Analyses of dietary intake were stratified by sex and age group. Results: The estimated overall prevalence of supplement use was 16.0% (95% CI: 15.4–16.6), with a higher prevalence among women (19.5% [95% CI: 18.7–20.5]) and the elderly (27.9% [95% CI: 26.4–29.4]). Women who used vitamin and/or mineral supplements showed higher average intakes for a greater number of dietary micronutrients compared to non-users. Conclusions: The findings from the analysis of average micronutrient intake from food sources, particularly among women and elderly women who used supplements, support the paradox of the “inverse supplement hypothesis”, which suggests that individuals who use dietary supplements are often those with the least need for them.

## 1. Introduction

Dietary supplements, also referred to as food supplements, are products designed to augment or complement the diet. They include a wide range of substances such as vitamins, minerals, plant-based compounds, botanical extracts, amino acids, and probiotics. These supplements are consumed orally and come in various forms, including tablets, capsules, powders, bars, gummies, and liquids [1]. Among the different types of dietary supplements, vitamin and/or mineral supplements are the most commonly used across populations in many countries [2,3].

Although supplements are intended to correct micronutrient deficiencies or maintain adequate intake, they are often classified as over-the-counter products and are frequently used by the adult population with the aim of improving overall health [4,5]. Studies investigating the use of these products among individuals over 20 years of age indicate that the highest prevalence of use is found among women and the elderly [2,6]. Comparisons between supplement users and non-users reveal that users tend to belong to higher social classes [7,8] and possess higher levels of education [3,9]. In addition to socioeconomic factors, supplement users are also more likely to engage in healthier lifestyle behaviors, such as greater physical activity [6,9,10], lower rates of smoking [5,10], and self-reporting excellent or very good health status [11]. However, concerns have arisen regarding the indiscriminate use of these products, as excessive intake of micronutrients may exceed tolerable upper intake levels, potentially leading to adverse reactions and toxicity [12]. In Brazil, the estimated prevalence of vitamin use in 2014 was 4.8% (95% CI: 4.3–5.3) among adults and the elderly, according to data from the National Survey on Access, Use, and Promotion of Rational Use of Medicines (PNAUM) [13].

The dietary habits of supplement users and non-users may also differ. Studies suggest that supplement users tend to consume more fruits and vegetables [14] as well as organic products [15] compared to non-users. A cohort study conducted in Germany, involving 13,615 women and 11,929 men aged 40 to 65 years, found that a higher intake of milk, dairy products, fish, and cereals was positively associated with the use of vitamin and/or mineral supplements. Additionally, this study observed that supplement use was linked to a lower consumption of meat and meat products, as well as a reduced intake of saturated fats [10].

Analyses of data from the National Health and Nutrition Examination Survey (NHANES) investigating vitamin and mineral intake from food among dietary supplement users and non-users demonstrated that micronutrient intake is higher among supplement users [16,17]. A study conducted in Brasília, Brazil, which aimed to evaluate the habitual intake of micronutrients from both food and dietary supplements, found that the average intake of micronutrients from food was higher among supplement users for the five micronutrients analyzed: calcium, folate, vitamin C, vitamin D, and vitamin E [18].

Data on the use of vitamin and/or mineral supplements in the general population, combined with dietary intake information, are highly relevant to public health. These products can play a critical role in addressing nutritional deficiencies, and their usage must be both effective and safe. In order to contribute to knowledge on how the pattern of use of these products is established in the Brazilian population and its impacts, the objective of this study is to estimate the prevalence of the use of vitamin and/or mineral supplements, as well as the dietary intake of micronutrients, among users and non-users in the adult and elderly population.

## 2. Methods

This study utilized data from the National Food Survey (INA), specifically from the personal food consumption module, which was applied to a subsample of households from the Household Budget Survey (POF), conducted by the Brazilian Institute of Geography and Statistics (IBGE) between 11 July 2017 and 9 July 2018.

The 2017–2018 POF employed a complex two-stage cluster sampling design, with geographic and statistical stratification of primary sampling units, corresponding to sectors or aggregates of sectors from the geographic base of the 2010 Demographic Census. The primary sampling units were selected using probability proportional to the number of households in each sector, within the final strata, which were chosen through simple random sampling. In the sampling plan adopted, permanent households, regarded as secondary sampling units, were selected by simple random sampling without replacement from the previously selected primary units.

To form the subsample for the individual food consumption block, it was determined that, due to the specific requirements of this instrument—such as the need for additional visits by the research agent—one in every three households selected for the overall sample would be randomly chosen to respond to questions regarding the individual food consumption of its residents. Consequently, this subsample was conducted across all sectors of the original survey sample. Data from the National Food Survey (INA) were collected from all residents aged 10 years or older in 20,112 selected households, representing a subsample of 34.7% of the original 2017–2018 Household Budget Survey (POF), which consisted of 57,920 households. As a result, information on the food consumption of 46,164 individuals was obtained.

Two 24 h dietary recalls (24hR) were administered on two non-consecutive days chosen throughout the week when the research agent was present in the selected household. The Automated Multiple-Pass Method [19] was used for data collection. In the first stage, trained interviewers prompted participants to recall everything they ate and drank on the day before the interview, recording a quick list on paper. In the second stage, data on portion sizes, preparation methods, and whether food was consumed at home or outside were entered into software developed specifically for the research using a tablet. The final stage involved estimating nutrient intake using version 7.0 of the Brazilian Food Composition Table (TBCA) from the Food Research Center of the University of São Paulo [20].

Methodological details regarding sampling design, data collection, training, and additional information are available in the IBGE’s official publications [21,22]. For this study, participants aged 20 years or older, who were non-pregnant, and who completed both 24 h recalls (*n* = 37,364) were included.

Participants were asked if they had used various supplements in the past 30 days, including multivitamins, multivitamin complexes, iron, ferrous sulfate, B vitamins, vitamin C, omega-3, fish oil, calcium, calcium with vitamin D, protein, creatine, other supplements for athletes, and other supplements. This study specifically investigated the use of vitamin and/or mineral supplements, defined as multivitamins, B vitamins, vitamin C, calcium, and iron. For analysis purposes, any participant who reported using at least one of these supplements was classified as a “user”.

Socioeconomic and demographic variables included sex, age group, household location (urban or rural), income class, and education level. In the analysis of micronutrient intake (calcium, magnesium, phosphorus, iron, copper, zinc, vitamin A, thiamine, riboflavin, niacin, cobalamin, pyridoxine, vitamin D, vitamin E, vitamin C, and folate) derived solely from food, the mean consumption over two 24 h recalls and corresponding 95% confidence intervals (95% CI) were used. Mean micronutrient consumption was compared against the Estimated Average Requirement (EAR) values proposed by the Institute of Medicine [23,24], which represent the estimated nutrient requirement by sex and age.

The prevalence of vitamin and/or mineral supplement use was calculated according to socioeconomic and demographic variables and expressed as relative frequency with 95% confidence intervals. Analyses accounted for the complex sampling design. A linear regression model was applied to test the statistical significance of differences in mean dietary intake between supplement users and non-users, with a significance level of 0.05. Statistical analyses were performed using PASW Statistics 18.0 for Windows (SPSS Inc., Chicago, IL, USA), employing the CSPLAN command to accommodate complex sampling designs.

## 3. Results

Among the 37,364 individuals included in this study, 5946 reported using vitamin and/or mineral supplements, representing 16.0% of the sample (95% CI: 15.4–16.6). The use of these supplements was higher among women (19.6%; 95% CI: 18.7–20.5), elderly individuals (27.9%; 95% CI: 26.4–29.4), urban residents (16.7%; 95% CI: 16.0–17.4), those in the highest income bracket (18.1%; 95% CI: 17.3–19.0), and those with a higher education degree (27.6%; 95% CI: 25.3–30.0). Multivitamins were the most commonly used supplements (7.0%; 95% CI: 6.6–7.4), followed by calcium (including calcium with vitamin D) at 5.4% (95% CI: 5.1–5.8) and vitamin C at 5.1% (95% CI: 4.7–5.5) (Table 1).

Among adult men, those who used vitamin and/or mineral supplements had a higher mean dietary intake of calcium and riboflavin compared to non-users. Conversely, for folate, the mean dietary intake was higher among non-users than among users. Elderly men who used supplements had a higher mean intake of vitamin C compared to non-users (Table 2). Despite supplementation, the average dietary intake of calcium, magnesium, vitamin A, pyridoxine, vitamin D, and vitamin E among both adult and elderly men—whether users or non-users—remained below the Estimated Average Requirement (EAR) for the population (Table 2).

Table 3 presents the average dietary intake of micronutrients among non-pregnant adult and elderly women. Adult women who used vitamin and/or mineral supplements had a higher average intake of calcium, phosphorus, riboflavin, pyridoxine, vitamin D, vitamin E, and vitamin C compared to non-users. In elderly women, supplement users had a higher average intake for ten of the sixteen micronutrients analyzed: calcium, phosphorus, iron, zinc, thiamine, riboflavin, pyridoxine, vitamin D, vitamin E, and vitamin C.

For adult women, the average intake of calcium, magnesium, vitamin A, pyridoxine, vitamin D, and vitamin E was below the EAR for both users and non-users of supplements. Similarly, among elderly women, the average intake of calcium, magnesium, pyridoxine, vitamin D, and vitamin E was below the EAR regardless of supplement use. Additionally, thiamine intake was lower than the recommended nutritional requirement in elderly women who did not use supplements (Table 3).

## 4. Discussion

The prevalence of vitamin and/or mineral supplement use in the study was 16.0% (95% CI 15.4–16.6), being higher in women, the elderly, and individuals with higher income and higher levels of education. The average dietary intakes of the micronutrients calcium, riboflavin, and vitamin C were significantly higher among users of vitamin and/or mineral supplements in both men and women. Among adult and elderly women, the average dietary intake of the micronutrients phosphorus, pyridoxine, vitamin D, and vitamin E was significantly higher among users of vitamin and/or mineral supplements compared to non-users.

Although the prevalence of vitamin and/or mineral supplement use is lower in Brazil when compared to other developed countries, the relationship between the age group with the highest use, income, and level of education is consistent with the literature [5,7,25]. The prevalence of micronutrient supplementation use increased from 46% to 49% in the United States between 2007 and 2018, progressively increasing with increasing age, income, and education, and is higher in women [11]. In Canada, the prevalence found was 31% in men aged 19 to 30 years and 67.8% in women over 71 years of age [26], and, in Australia, the prevalence of multivitamin and/or multimineral use in the population aged 30 to 49 years was 22.3% [27].

The higher prevalence of vitamin and/or mineral use among women and the elderly, as identified in this study, aligns with findings from a previous analysis of supplement use based on data from another Brazilian survey [13]. Numerous studies on the prevalence of supplement use have also consistently shown higher usage rates in these demographic groups [5,7], further reinforcing this observation with substantial evidence from the literature.

In the evaluation of dietary intake among users and non-users of vitamin and/or mineral supplements, it was observed that the average consumption of micronutrients ingested through food was higher for ten different micronutrients analyzed among women who used these supplements when compared to non-users, especially among elderly Brazilian women. When men were analyzed, no significant differences were observed in the average dietary intake of micronutrients between users and non-users of vitamin and/or mineral supplements. However, differences were found for calcium, riboflavin, and vitamin C, with higher average intakes among users compared to non-users of vitamin and/or mineral supplements. This finding was consistent with a study that used data from the Korea National Health and Nutrition Examination Survey IV (KNHANES) conducted between 2007 and 2009 [3]. Another study, using data from the NHANES 2003–2006 in the United States of America, found that, for women, there was a higher average usual dietary intake of eight minerals in users compared to non-users of dietary supplements for the selected dietary minerals, except for selenium [16]. When studies have analyzed micronutrient supplement users with respect to their health, they have typically not been found to have deficiencies but rather have been found to make healthier dietary choices and have a healthier lifestyle than non-users, including a tendency to devote more time to physical activity than non-users [10,15]. Indeed, there appears to be a disparity between nutritional and health needs and dietary supplement use. Most supplement users use them for preventive purposes and are more health-conscious than non-users of supplements [3,4,5], although it has been shown that supplementing the diet of well-nourished adults with mineral or vitamin supplements provides no clear benefit and may even be harmful, so vitamins should not be used for chronic disease prevention in this context [28]. Those who take supplements for treatment purposes may have underlying health indications and may be more likely to benefit from supplementation than those who take supplements for preventive purposes [29]. This paradox forms the basis for the inverse supplement hypothesis, that is, people who are most likely to use dietary supplements appear to be those who need them the least [30].

The inverse supplement hypothesis was first introduced by Kirk, Woodhouse, and Conner [30], who referenced the study by Draper et al. [31], which evaluated energy and nutrient intake in three groups of vegetarians (semi-vegetarians, ovo-lacto-vegetarians, and vegans). Subsequently, Kirk et al. [32] used data from the United Kingdom Women’s Cohort Study, involving 13,822 women, to test this hypothesis. They concluded that supplement use is associated with a healthier lifestyle and adequate nutritional intake, suggesting that supplement users may not require supplements to address nutritional deficiencies. Other studies have also explored the inverse supplement hypothesis in efforts to confirm or refute it [29,33,34]. In this analysis, the results obtained point towards supporting the inverse supplement hypothesis in women, especially among elderly women.

This study also demonstrates that, within this representative sample of Brazilian adults, the average intake of calcium, magnesium, and vitamins B6, D, and E falls below the Estimated Average Requirements (EAR) for both men and women, regardless of their use of vitamin and/or mineral supplements. Only in the case of vitamin A did the average dietary intake of women—both adult and elderly—who used supplements meet the EAR. Inadequate micronutrient intake remains a public health concern and can be attributed to a complex web of determinants, including food insecurity, insufficient consumption of natural and minimally processed foods, and poor overall diet quality [35].

In Brazil, the National Food Survey (INA) highlighted the issue of micronutrient deficiencies in the population [22]. As such, strategies aimed at promoting access to healthy foods are essential for reducing these deficiencies. The decline in the consumption of ultra-processed foods among the highest-income population, first observed in the 2017–2018 Household Budget Survey (POF), may indicate a shift in social norms regarding the consumption of such foods [36]. This higher-income group also exhibited a higher prevalence of vitamin and/or mineral supplement use, as noted in this study.

Several limitations should be considered. As the data are only for Brazilian adults and the elderly, the results may not be extended to populations in other countries. Food consumption data were obtained through 24 h recalls (24hR), a widely used tool in epidemiological studies. While effective, 24hR may influence the accuracy of findings because they rely on recent memory and tend to underestimate intake, particularly among elderly individuals and children [37]. The lower reported values for dietary micronutrients may, in part, be attributed to the underreporting of foods consumed less frequently, which may serve as primary sources of certain micronutrients. Despite these limitations, the 24hR method is one of the most commonly used due to its high precision and response rate. This method is systematically applied in national surveys, such as the National Health and Nutrition Examination Survey (NHANES) in the United States, which employs a similar approach to that of this study, including the use of the multiple-pass method and specialized software for data collection.

## 5. Conclusions

In conclusion, the prevalence of vitamin and/or mineral supplement use among Brazilian adults and the elderly was higher among women, the elderly, those with higher education levels, and individuals in higher economic classes. The analysis of dietary micronutrient intake revealed that women who use vitamin and/or mineral supplements, especially elderly women, consume higher levels of these nutrients from food than those who do not use supplements. This finding aligns with the “inverse supplement hypothesis”, which suggests that individuals who use dietary supplements are often those who need them the least.

The typical Brazilian diet is influenced by a range of cultural and socioeconomic factors, with income disparities clearly reflected in food consumption patterns across different social strata. Information on the use of vitamin and/or mineral supplements, alongside comparisons of micronutrient intake from food sources, can contribute to discussions within the framework of the National Food and Nutrition Policy [38]. Given that dietary supplements are part of the nutritional practices of a portion of the population and serve as important sources of micronutrients, their inclusion in nutritional assessments is essential.

The implementation of a large-scale food survey to assess individual consumption across different regions, sexes, and age groups represents a significant advancement for research in the field of nutritional epidemiology in Brazil. The inclusion of dietary supplement use in the second edition of the survey, INA 2017–2018, marks the first time these data have been collected on a national scale, offering valuable insights into the consumption of these products. It is expected that knowledge can advance in new investigations of these products in the Brazilian population, thus enabling the construction of trends over time.

## Figures and Tables

**Table 1 nutrients-16-03815-t001:** Prevalence of vitamin and/or mineral supplement use and corresponding 95% confidence intervals (CI) among the Brazilian population aged 20 years and older, according to socioeconomic and demographic data. National Household Budget Survey/National Food Survey 2017–2018 (POF/INA 2017–2018) (*n* = 37,364).

Variables	Vitamin and/or Mineral Supplement ^a^			Multivitamins			Vitamin C			Vitamin B Complex			Calcium ^b^			Iron		
	%	95% CI	*p*-Value *	%	95% CI	*p*-Value *	%	95% CI	*p*-Value *	%	95% CI	*p*-Value *	%	95% CI	*p*-Value *	%	95% CI	*p*-Value *
Sex																		
Men	12.1	11.3–12.9	<0.001	5.9	5.3–6.5	<0.001	4.7	4.2–5.3	<0.001	2.0	1.7–2.4	<0.001	2.7	2.4–3.1	<0.001	1.3	1.0–1.6	<0.001
Women	19.6	18.7–20.5		8.0	7.4–8.6		5.4	5.0–6.0		3.2	2.8–3.6		7.9	7.3–8.5		3.0	2.7–3.3	
Age group (years)																		
20–59	12.7	12.1–13.4	<0.001	5.5	5.1–6.0	<0.001	5.1	4.6–5.5	0.030	2.3	2.0–2.6	0.037	2.9	2.6–3.2	<0.001	2.0	1.7–2.2	<0.001
60 and over	27.9	26.4–29.4		12.2	11.2–13.3		5.2	4.5–5.9		4.1	3.5–4.7		14.8	13.6–16.0		2.9	2.3–3.5	
Region																		
North	19.6	17.5–21.9	0.026	8.3	6.7–10.1	0.009	10.1	8.6–11.8	0.001	3.5	2.7–4.7	0.012	5.4	4.3–6.7	0.023	3.4	2.6–4.4	<0.001
Northeast	16.0	15.2–16.9		6.6	6.1–7.2		5.7	5.2–6.3		2.4	2.1–2.8		5.3	4.8–5.8		2.2	1.9–2.6	
Southeast	15.4	14.4–16.6		6.8	6.1–7.5		4.2	3.6–5.0		2.8	2.3–3.3		5.3	4.7–6.0		1.9	1.5–2.4	
South	14.6	13.4–15.9		6.5	5.6–7.4		3.1	2.5–3.8		2.3	1.8–2.8		6.3	5.6–7.2		2.0	1.6–2.5	
Midwest	18.4	16.6–20.4		9.1	8.0–10.4		6.8	5.4–8.4		2.5	1.9–3.2		5.1	4.3–6.0		2.7	2.1–3.4	
Housing situation																		
Urban	16.7	16.0–17.4	0.408	7.3	6.8–7.7	0.003	5.4	5.0–5.9	0.014	2.8	2.5–3.1	0.709	5.6	5.2–6.0	0.028	2.2	2.0–2.5	0.001
Rural	11.9	11.0–12.8		5.2	4.6–5.8		2.9	2.5–3.4		1.9	1.6–2.3		4.2	3.7–4.8		2.0	1.6–2.4	
Income per month ^c^ (mw = minimum wage)																		
Up to 1 mw	7.9	6.5–9.5	0.772	3.3	2.3–4.6	0.024	2.1	1.5–2.8	0.013	1.2	0.7–2.0	0.042	1.6	1.1–2.2	0.166	1.7	1.2–2.5	0.345
1 to 3 mw	13.4	12.4–14.1		5.4	4.8–6.0		3.7	3.3–4.2		2.2	1.9–2.7		4.6	4.1–5.1		2.2	1.8–2.7	
Over 3 mw	18.1	17.3–19.0		8.1	7.5–8.7		6.0	5.5–6.6		3.0	2.6–3.4		6.2	5.7–6.7		2.2	1.9–2.5	
Education level																		
No or incomplete elementary	14.4	13.6–15.2	0.044	5.9	5.4–6.4	<0.001	3.1	2.8–3.5	0.001	2.1	1.9–2.5	<0.001	5.7	5.2–6.2	0.015	2.2	1.9–2.6	0.028
Complete elementary	13.4	11.9–15.0		4.9	4.1–5.9		4.9	3.9–6.2		1.9	1.5–2.6		4.7	3.9–5.8		2.2	1.6–2.8	
Complete high school	14.4	13.4–15.5		6.2	5.5–7.0		5.6	4.9–6.4		3.1	2.5–3.8		4.1	3.6–4.8		2.0	1.6–2.6	
Complete higher	27.6	25.3–30.0		14.2	12.6–16.0		10.7	9.2–12.5		4.0	3.3–4.9		7.9	6.7–9.3		2.4	1.9–3.0	
Total	16.0	15.4–16.6		7.0	6.6–7.4		5.1	4.7–5.5		2.7	2.4–2.9		5.4	5.1–5.8		2.2	1.9–2.4	

***** Chi square test. ^a^ Corresponds to the sum of multivitamins, vitamin C, B vitamins, calcium (including calcium + vitamin D), and iron. ^b^ Includes products containing both calcium and vitamin D. ^c^ The reference value of BRL 954.00 (nine hundred and fifty-four reais) was used, based on the minimum wage in effect on 15 January 2018, the survey’s reference date. Source: Prepared by the author based on microdata from the Household Budget Survey/National Food Survey 2017–2018 (POF/INA 2017–2018).

**Table 2 nutrients-16-03815-t002:** Mean intake of micronutrients from foods and their corresponding 95% confidence intervals (CI) among adult men (20–59 years old) (*n* = 13,338) and elderly men (60 years and older) (*n* = 3789), as obtained through 24 h dietary recalls (R24h), categorized by users and non-users of vitamin and/or mineral supplements. National Household Budget Survey/National Food Survey 2017–2018 (POF/INA 2017–2018).

				Vitamin and/or Mineral Supplements				
		Distribution		Users		Non-Users		
Micronutrients	EAR ^a^	Mean	95% CI	Mean	95% CI	Mean	95% CI	*p*-Value *
	Adult Men							
Calcium, mg	800	475.3	465.1–485.5	534.7	500.5–568.8	468.6	457.9–479.2	<0.0001 **
Magnesium, mg	330/350 ^1^	308	304.3–311.8	299.2	285.8–312.7	309	305.2–312.9	0.109
Phosphorus, mg	580	1160.4	1146.5–1174.4	1178.1	1133.2–1222.9	1158.5	1143.7–1173.2	0.416
Iron, mg	6	12.8	12.7–13	12.7	12.3–13.2	12.8	12.7–13	0.686
Copper, mg	0.7	1.6	1.6–1.7	1.5	1.4–1.6	1.6	1.6–1.7	0.112
Zinc, mg	9.4	12.8	12.6–12.9	12.1	11.6–12.7	12.8	12.3–13	0.023
Vitamin A, mcg ^b^	625	430.1	393.5–466.8	462	375.1–548.8	426.5	386.9–466.2	0.467
Thiamine, mg	1	1.1	1.1–1.1	1.1	1–1.1	1.1	1.1–1.1	0.933
Riboflavin, mg	1.1	1.1	1.1–1.1	1.2	1.2–1.3	1.1	1.1–1.1	0.001
Niacin, mg ^c^	12	20.3	19.8–20.8	20.5	19–21.9	20.2	19.7–20.8	0.757
Cobalamin, mcg	2	4.9	4.7–5	5	4.5–5.4	4.9	4.7–5	0.682
Pyridoxine, mg	1.1/1.4 ^2^	0.8	0.8–0.9	0.9	0.8–0.9	0.8	0.8–0.9	0.444
Vitamin D (cholecalciferol), mcg	10	1.5	1.5–1.6	1.7	1.5–1.9	1.5	1.5–1.6	0.029
Vitamin E (alpha-tocopherol), mg	12	7.3	7.2–7.5	7.8	7.4–8.3	7.3	7.1–7.4	0.030
Vitamin C, mg	75	121.8	116.6–126.9	128.4	112.1–144.6	121	115.6–126.4	0.403
Folate, mcg ^d^	320	479.7	473.1–486.4	458.5	436.9–480.2	482.2	475.2–489.1	0.041
	Elderly men							
Calcium, mg	800/1000 ^3^	450	432.4–467.7	471	440–501.9	445	424.4–465.6	0.171
Magnesium, mg	350	275.6	270.3–280.9	273.6	260.9–286.4	276.1	270.2–281.9	0.731
Phosphorus, mg	580	979.5	957.3–1001.6	985.5	936–1035	978	953.2–1002.8	0.790
Iron, mg	6	10.9	10.6–11	10.8	10.2–11.4	10.9	10.6–11.2	0.690
Copper, mg	0.7	1.4	1.4–1.5	1.5	1.4–1.6	1.4	1.4–1.5	0.275
Zinc, mg	9.4	11	10.7–11.3	10.8	10.1–11.4	11	10.7–11.4	0.461
Vitamin A, mcg ^b^	625	383	336.7–429.3	475.3	338.7–611.9	360.7	313.6–407.8	0.120
Thiamine, mg	1	0.9	0.9–1	1	0.9–1	0.9	0.9–1	0.368
Riboflavin, mg	1.1	1.1	1.0–1.1	1.1	1–1.2	1.1	1.0–1.1	0.507
Niacin, mg ^c^	12	16	15.3–16.7	15.5	14.2–16.7	16.1	15.3–16.9	0.411
Cobalamin, mcg	2	3.9	3.7–4.1	4.2	3.5–4.8	3.9	3.6–4.1	0.342
Pyridoxine, mg	1.4	0.7	0.7–0.7	0.7	0.6–0.7	0.7	0.7–0.7	0.663
Vitamin D (cholecalciferol), mcg	10	1.4	1.3–1.5	1.5	1.3–1.7	1.4	1.3–1.5	0.340
Vitamin E (alpha-tocopherol), mg	12	6.4	6.2–6.7	6.5	6.0–6.9	6.4	6.1–6.7	0.808
Vitamin C, mg	75	109.5	102–116.9	142.1	115.9–168.4	101.6	95.1–108.1	0.003
Folate, mcg ^d^	320	412.7	403–422.4	411.5	390.1–432.9	413	402.1–423.9	0.901

* Wald test. ** *p* < 0.01. ^a^ EAR: Estimated Average Requirement, reference values proposed by the Institute of Medicine. ^b^ Retinol Activity Equivalent (RAE). ^c^ Preformed niacin. ^d^ Dietary Folate Equivalent. ^1^ EAR: 330 for ages 20–30/EAR: 350 for ages 31–59. ^2^ EAR: 1.1 for ages 20–50/EAR: 1.4 for ages 51–59. ^3^ EAR: 800 for ages 60–70/EAR: 1000 for ages 71 and older. Source: Prepared by the authors based on microdata from the Household Budget Survey/National Food Survey 2017–2018 (POF/INA 2017–2018).

**Table 3 nutrients-16-03815-t003:** Mean intake of micronutrients from food and their corresponding 95% confidence intervals (CI) among adult women (20–59 years old) (*n* = 15,694) and elderly women (60 years and older) (*n* = 4543) who were non-pregnant, as obtained through 24 h dietary recalls, categorized by users and non-users of vitamin and/or mineral supplements. National Household Budget Survey/National Food Survey 2017–2018 (POF/INA 2017–2018).

				Vitamin and/or Mineral Supplements				
		Distribution		Users		Non-Users		
Micronutrients	EAR ^a^	Mean	95% CI	Mean	95% CI	Mean	95% CI	*p*-Value *
	Adult women							
Calcium, mg	800/1000 ^1^	403.6	395.9–411.3	473.3	451.0–495.5	391.2	383.5–399.2	<0.0001 **
Magnesium, mg	255/265 ^2^	232.7	230.1–235.3	238.9	232.3–245.5	231.6	228.8–234.4	0.045
Phosphorus, mg	580	877.4	868.0–886.9	919.6	890.7–942.5	870.5	860.4–880.6	0.001
Iron, mg	8.1/5 ^3^	9.6	9.5–9.7	9.5	9.3–9.8	9.6	9.4–9.7	0.915
Copper, mg	0.7	1.3	1.2–1.3	1.3	1.2–1.4	1.3	1.2–1.3	0.283
Zinc, mg	6.8	9.4	9.3–9.6	9.6	9.3–9.9	9.4	9.3–9.6	0.252
Vitamin A, mcg ^b^	500	427.7	388.2–467.3	490.4	380.3–600.5	416.6	374.3–458.8	0.220
Thiamine, mg	0.9	0.9	0.8–0.9	0.9	0.8–0.9	0.9	0.8–0.9	0.218
Riboflavin, mg	0.9	0.9	0.9–1.0	1	1–1.1	0.9	0.9–0.9	<0.0001 **
Niacin, mg ^c^	11	14.7	14.4–15	15.2	14.5–16.0	14.6	14.2–14.9	0.121
Cobalamin, mcg	2	3.8	3.6–4	4.2	3.7–4.6	3.8	3.6–4.0	0.192
Pyridoxine, mg	1.1/1.3 ^4^	0.6	0.6–0.6	0.6	0.6–0.7	0.6	0.6–0.6	0.028
Vitamin D (cholecalciferol), mcg	10	1.3	1.2–1.3	1.5	1.4–1.6	1.2	1.2–1.3	<0.0001 **
Vitamin E (alpha-tocopherol), mg	12	6	5.9–6.2	6.8	6.4–7.2	5.9	5.8–6	<0.0001 **
Vitamin C, mg	60	112	108.4–115.6	129.9	119.8–140	108.8	105–112.6	<0.0001 **
Folate, mcg ^d^	320	353.8	349.4–358.2	345.7	335.4–356	355.2	350.4–360.1	0.101
	Elderly women							
Calcium, mg	1000	423.3	410.7–435.9	473.7	451.2–496.31	396.6	381.6–411.6	<0.0001 **
Magnesium, mg	265	221.7	217.7–225.7	226.4	220–232.9	219.1	214.0–224.2	0.081
Phosphorus, mg	580	799.3	784.9–813.6	835.5	813.2–857.7	780.1	761.8–798.5	<0.0001 **
Iron, mg	5	8.6	8.4–8.8	8.9	8.6–9.3	8.4	8.2–8.6	0.004
Copper, mg	0.7	1.3	1.2–1.3	1.3	1.2–1.3	1.3	1.1–1.4	0.914
Zinc, mg	6.8	8.4	8.3–8.6	8.8	8.4–9.1	8.3	8–8.5	0.026
Vitamin A, mcg ^b^	500	492.8	406.7–578.9	458.7	387–530.5	510.8	384.8–636.9	0.481
Thiamine, mg	0.9	0.8	0.8–0.8	0.9	0.8–0.9	0.8	0.8–0.8	0.004
Riboflavin, mg	0.9	1	1–1	1.1	1.0–1.1	1	0.9–1	0.010
Niacin, mg ^c^	11	13	12.5–13.4	13.5	12.8–14.3	12.7	12.1–13.2	0.071
Cobalamin, mcg	2	3.5	3.2–3.9	3.4	3.1–3.7	3.6	3.1–4.2	0.494
Pyridoxine, mg	1.3	0.5	0.5–0.5	0.6	0.5–0.6	0.5	0.5–0.5	0.003
Vitamin D (cholecalciferol), mcg	10	1.2	1.1–1.3	1.3	1.2–1.5	1.1	1.1–1.2	0.012
Vitamin E (alpha-tocopherol), mg	12	5.4	5.2–5.6	5.7	5.4–6	5.2	5–5.5	0.016
Vitamin C, mg	60	109.7	103.7–115.6	127.6	117.2–138.1	100.2	93–107.4	<0.0001 **
Folate, mcg ^d^	320	329	321.8–336.1	336.8	325.7–347.8	324.8	315.7–334	0.103

* Wald test. ** *p* < 0.01. ^a^ EAR: Estimated Average Requirement, reference values proposed by the Institute of Medicine. ^b^ Retinol Activity Equivalent (RAE). ^c^ Preformed niacin. ^d^ Dietary Folate Equivalent. ^1^ EAR: 800 for ages 20–50/EAR: 1000 for ages 51–59. ^2^ EAR: 255 for ages 20–50/EAR: 265 for ages 51–59. ^3^ EAR: 8.1 for ages 20–50/EAR: 5 for ages 51–59. ^4^ EAR: 1.1 for ages 20–50/EAR: 1.3 for ages 51–59. Source: Prepared by the authors based on microdata from the Household Budget Survey/National Food Survey 2017–2018 (POF/INA 2017–2018).

## Data Availability

Data supporting reported results can be found at https://ftp.ibge.gov.br/Orcamentos_Familiares/Pesquisa_de_Orcamentos_Familiares_2017_2018/Microdados/ (accessed on 5 November 2024).

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
