# Peer review of "Dietary Intake of Micronutrients and Use of Vitamin and/or Mineral Supplements: Brazilian National Food Survey"

_nutrients, 2024, doi:10.3390/nu16223815_

Round 1
Reviewer 1 Report
Comments and Suggestions for Authors
Review nutrients-3237786-v1
The work contains some useful information. However, the overall novelty of this manuscript was not significant enough. There are some similar reports with similar strategies but from different nations/countries.
For the current data reported, as all from survey and recall etc, how to validate or verify the conclusions? Is that possible to select some parameters with certain number of population to run further analysis (lab or other calculation) to make the conclusion more solid?
Author Response
Dear Prof. Dr. Lluis Serra-Majem
Editor-in-Chief, Nutrients
Thank you for the thoughtful review, it helped improve the manuscript. We have responded to reviewers’ comments and revised the manuscript, accordingly, indicating in red the changes we made.
Reviewer: 1
Comments to the Author
The work contains some useful information. However, the overall novelty of this manuscript was not significant enough. There are some similar reports with similar strategies but from different nations/countries.
For the current data reported, as all from survey and recall etc, how to validate or verify the conclusions? Is that possible to select some parameters with certain number of population to run further analysis (lab or other calculation) to make the conclusion more solid?
Response: Thank you for this comment. Our findings have scientific relevance for Brazilian public health and can be compared with other populations, thus being relevant for global public health. The data for this study were obtained from the National Food Survey (INA), a component of the Household Budget Survey (POF), which is one of the most important and comprehensive datasets on food consumption in Brazil. As part of the POF, the INA provides extensive coverage and includes a representative sample of the entire Brazilian population. It plays a crucial role in identifying food consumption patterns, trends, and regional or socioeconomic disparities across the country. The data collected are widely utilized to inform public policies, such as the Food Guide for the Brazilian Population, as well as to monitor nutritional risk factors for chronic diseases. The survey employs the 24-hour food recall method (R24h), administered on two non-consecutive days, enabling a more detailed analysis of dietary habits. However, the dataset does not include laboratory test results. Access to the database was made available to allow other researchers to validate the data and conduct further analyses. Certain aspects of the conclusions were revised to enhance their robustness.
Reviewer 2 Report
Comments and Suggestions for Authors
In this descriptive study, the authors examined the dietary intake of different micronutrients among Brazilian subjects who used or did not use mineral and/or vitamin supplements. Some differences in micronutrient intake were observed between supplement users/non-users, men/women, adults/elderly. The results are interesting, well presented and discussed. Here are some suggestions to improve the quality of the manuscript:
-I think is necessary to be better present the Tables (format) and in particular the footnotes which are not easily readable since several symbols (letter, number and *) are used and sometime are missing.
-The results reported in Table 1 should be statistically analysed and the significant p values reported
Author Response
Comments to the Author
In this descriptive study, the authors examined the dietary intake of different micronutrients among Brazilian subjects who used or did not use mineral and/or vitamin supplements. Some differences in micronutrient intake were observed between supplement users/non-users, men/women, adults/elderly. The results are interesting, well presented and discussed. Here are some suggestions to improve the quality of the manuscript:
-I think is necessary to be better present the Tables (format) and in particular the footnotes which are not easily readable since several symbols (letter, number and *) are used and sometime are missing.
-The results reported in Table 1 should be statistically analysed and the significant p values reported
Response: Thank you for the careful review. We have revised the tables to make them easier to read. We have added the p-value to Table 1 as suggested.
Reviewer 3 Report
Comments and Suggestions for Authors
This study examines the prevalence of vitamin and/or mineral supplement use and assesses the dietary intake of micronutrients among adult and elderly populations in Brazil, utilizing data from the 2017–2018 Brazilian National Food Survey. The research finds that 16% of the population uses supplements, with higher usage among women and the elderly, and suggests that individuals who use supplements often already meet their micronutrient needs through diet, supporting the "inverse supplement hypothesis." The authors analyzed dietary intake across various micronutrients and stratified the data by sex and age group.
This is a very interesting and well-written study, and I want to congratulate the authors on their thorough work. However, it’s important to note that the results may not be generalizable to other populations, as the study focused solely on a South American sample. Despite this limitation, the research is excellent, and the findings are both valuable and timely. In future studies using this model, I suggest the authors consider examining the impact of herbal supplements and their association with liver injury, as this issue is becoming increasingly common. Well done!
Comments on the Quality of English Language
Minor editing of English language required.
Author Response
Dear Prof. Dr. Lluis Serra-Majem
Editor-in-Chief, Nutrients
Thank you for the thoughtful review, it helped improve the manuscript. We have responded to reviewers’ comments and revised the manuscript, accordingly, indicating in red the changes we made.
Reviewer: 1
Comments to the Author
The work contains some useful information. However, the overall novelty of this manuscript was not significant enough for being published in this current journal. There are some similar reports with similar strategies but from different nations/countries.
For the current data reported, as all from survey and recall etc, how to validate or verify the conclusions? Is that possible to select some parameters with certain number of population to run further analysis (lab or other calculation) to make the conclusion more solid?
Response: Thank you for this comment. Our findings have scientific relevance for Brazilian public health and can be compared with other populations, thus being relevant for global public health. The data for this study were obtained from the National Food Survey (INA), a component of the Household Budget Survey (POF), which is one of the most important and comprehensive datasets on food consumption in Brazil. As part of the POF, the INA provides extensive coverage and includes a representative sample of the entire Brazilian population. It plays a crucial role in identifying food consumption patterns, trends, and regional or socioeconomic disparities across the country. The data collected are widely utilized to inform public policies, such as the Food Guide for the Brazilian Population, as well as to monitor nutritional risk factors for chronic diseases. The survey employs the 24-hour food recall method (R24h), administered on two non-consecutive days, enabling a more detailed analysis of dietary habits. However, the dataset does not include laboratory test results. Access to the database was made available to allow other researchers to validate the data and conduct further analyses. Certain aspects of the conclusions were revised to enhance their robustness.
Reviewer: 2
Comments to the Author
In this descriptive study, the authors examined the dietary intake of different micronutrients among Brazilian subjects who used or did not use mineral and/or vitamin supplements. Some differences in micronutrient intake were observed between supplement users/non-users, men/women, adults/elderly. The results are interesting, well presented and discussed. Here are some suggestions to improve the quality of the manuscript:
-I think is necessary to be better present the Tables (format) and in particular the footnotes which are not easily readable since several symbols (letter, number and *) are used and sometime are missing.
-The results reported in Table 1 should be statistically analysed and the significant p values reported
Response: Thank you for the careful review. We have revised the tables to make them easier to read. We have added the p-value to Table 1 as suggested.
Reviewer 3:
Comments to the Author
This study examines the prevalence of vitamin and/or mineral supplement use and assesses the dietary intake of micronutrients among adult and elderly populations in Brazil, utilizing data from the 2017–2018 Brazilian National Food Survey. The research finds that 16% of the population uses supplements, with higher usage among women and the elderly, and suggests that individuals who use supplements often already meet their micronutrient needs through diet, supporting the "inverse supplement hypothesis." The authors analyzed dietary intake across various micronutrients and stratified the data by sex and age group.
This is a very interesting and well-written study, and I want to congratulate the authors on their thorough work. However, it’s important to note that the results may not be generalizable to other populations, as the study focused solely on a South American sample. Despite this limitation, the research is excellent, and the findings are both valuable and timely. In future studies using this model, I suggest the authors consider examining the impact of herbal supplements and their association with liver injury, as this issue is becoming increasingly common. Well done!
Response: Thank you for your review and suggestions for future studies, which will certainly be considered.
In the review, we added that the findings are limited to the population under study, as suggested by you (lines 291-292).
Reviewer 4:
Comments to the Author:
Pavlak and colleagues present an interesting study on the prevalence of vitamin and mineral supplement use among Brazilian adults and the elderly.
Response: Thank you for the careful review. We will respond to the points raised in your comments. You have certainly added valuable contributions.
The objectives of the study are stated in a very broad way: "estimate the prevalence of vitamin and/or mineral supplement use and assess the dietary intake of micronutrients among users and non-users." However, it is unclear what specific hypotheses the study aims to test. For example, does it seek to explore the health impacts of supplement use, or is it simply descriptive? More clarity in the objective would make the study’s purpose more distinct.
Response: We appreciate the comment and we had reviewed the objectives carefully for improvement. Please, see the topic in the revised version of the manuscript (lines 75-79).
Also, a more detailed breakdown of how the different age groups and sexes differ in terms of both supplements use and dietary intake would add depth to the analysis.
Response: Thank you for this suggestion. We have included more information about this in lines 235-239.
The "inverse supplement hypothesis" is briefly mentioned, however, it is not fully explained or critically discussed. Why might individuals with higher dietary intake of micronutrients still feel the need to take supplements? A deeper examination of possible psychological, social, or economic factors contributing to this phenomenon would enhance the reader’s understanding.
Response: Following the reviewer suggestion, we added a more detailed explanation of the aforementioned hypothesis to facilitate the reader's understanding. (lines 270-277).
Women, particularly elderly women, have higher average intakes of micronutrients compared to non-users. However, there is no comparable analysis of men or younger populations. This gender-specific focus is not well explained.
Response: Good point. Men, adults and elderly, do not present significant statistical differences in the analyses carried out. We tried to clarify in the text (lines 237-241).
Were there any limitations in the self-reported data that could affect accuracy?
Response: We have taken care to clarify the limitations of the method and its biases. Please see the included sentence below:
Lines 300-303:
“While effective, 24hR may influence the accuracy of findings because it relies on recent memory and tends to underestimate intake, particularly among elderly individuals and children [37]. The lower reported values for dietary micronutrients may, in part, be attributed to the underreporting of foods consumed less frequently, which may serve as primary sources of certain micronutrients”.
Additionally, it would be useful to know how the dietary intake of micronutrients was adjusted for factors like total calorie intake or food group consumption.
Response: Good point. Given that the objective of this manuscript is to estimate the prevalence of dietary micronutrient intake among participants who do or do not use dietary supplements, diet is not considered in this study as an exposure factor of interest that may be potentially associated with a health/disease outcome. The primary goal of adjusting nutrient and food intake for total dietary energy in epidemiological studies is to assess how variations in diet composition impact health outcomes. In this sense, we follow descriptive methodologies similar to those employed in NHANES and the Household Budget Survey (POF), which are discussed in the manuscript, and present the unadjusted mean values ​​of micronutrient intake estimated from the two 24-hour dietary recalls (24hR).
The study does not suggest public health considerations and which efforts should be made to regulate supplement use, particularly among individuals who already meet their nutritional needs through diet? A discussion on the policy implications could enhance the relevance of the study’s conclusions.
Response: In lines 290-297 we address important topics for knowledge about the diet of the Brazilian population. We modified the conclusion to make the contributions to Brazilian public health more specific.
The conclusion seems repetitive, reiterating the “inverse supplement hypothesis” without offering new insights or a strong conclusion on how this finding should be interpreted in terms of action or further research. The ending could be more impactful by proposing future directions for research or policy recommendations based on the findings.
Response: Thank you for the careful review. We revised the conclusion according to your suggestion.
Reviewer 4 Report
Comments and Suggestions for Authors
Pavlak and colleagues present an interesting study on the prevalence of vitamin and mineral supplement use among Brazilian adults and the elderly.
The objectives of the study are stated in a very broad way: "estimate the prevalence of vitamin and/or mineral supplement use and assess the dietary intake of micronutrients among users and non-users." However, it is unclear what specific hypotheses the study aims to test. For example, does it seek to explore the health impacts of supplement use, or is it simply descriptive? More clarity in the objective would make the study’s purpose more distinct.
Also, a more detailed breakdown of how the different age groups and sexes differ in terms of both supplements use and dietary intake would add depth to the analysis.
The "inverse supplement hypothesis" is briefly mentioned, however, it is not fully explained or critically discussed. Why might individuals with higher dietary intake of micronutrients still feel the need to take supplements? A deeper examination of possible psychological, social, or economic factors contributing to this phenomenon would enhance the reader’s understanding.
Women, particularly elderly women, have higher average intakes of micronutrients compared to non-users. However, there is no comparable analysis of men or younger populations. This gender-specific focus is not well explained.
Were there any limitations in the self-reported data that could affect accuracy? Additionally, it would be useful to know how the dietary intake of micronutrients was adjusted for factors like total calorie intake or food group consumption.
The study does not suggest public health considerations and which efforts should be made to regulate supplement use, particularly among individuals who already meet their nutritional needs through diet? A discussion on the policy implications could enhance the relevance of the study’s conclusions.
The conclusion seems repetitive, reiterating the “inverse supplement hypothesis” without offering new insights or a strong conclusion on how this finding should be interpreted in terms of action or further research. The ending could be more impactful by proposing future directions for research or policy recommendations based on the findings.
Author Response
Comments to the Author:
Pavlak and colleagues present an interesting study on the prevalence of vitamin and mineral supplement use among Brazilian adults and the elderly.
Response: Thank you for the careful review. We will respond to the points raised in your comments. You have certainly added valuable contributions.
The objectives of the study are stated in a very broad way: "estimate the prevalence of vitamin and/or mineral supplement use and assess the dietary intake of micronutrients among users and non-users." However, it is unclear what specific hypotheses the study aims to test. For example, does it seek to explore the health impacts of supplement use, or is it simply descriptive? More clarity in the objective would make the study’s purpose more distinct.
Response: We appreciate the comment and we had reviewed the objectives carefully for improvement. Please, see the topic in the revised version of the manuscript (lines 75-79).
Also, a more detailed breakdown of how the different age groups and sexes differ in terms of both supplements use and dietary intake would add depth to the analysis.
Response: Thank you for this suggestion. We have included more information about this in lines 230-235
The "inverse supplement hypothesis" is briefly mentioned, however, it is not fully explained or critically discussed. Why might individuals with higher dietary intake of micronutrients still feel the need to take supplements? A deeper examination of possible psychological, social, or economic factors contributing to this phenomenon would enhance the reader’s understanding.
Response: In accordance with the reviewer suggestion, we added a more detailed explanation of the aforementioned hypothesis to facilitate the reader's understanding. (lines 265-273).
Women, particularly elderly women, have higher average intakes of micronutrients compared to non-users. However, there is no comparable analysis of men or younger populations. This gender-specific focus is not well explained.
Response:: Good point. Men, adults and elderly, do not present significant statistical differences in the analyses carried out. We tried to clarify in the text (line 240-244).
Were there any limitations in the self-reported data that could affect accuracy?
Response: We have taken care to clarify the limitations of the method and its biases. Please see the included sentence below:
Lines 296-298:
“While effective, 24hR may influence the accuracy of findings because it relies on recent memory and tends to underestimate intake, particularly among elderly individuals and children [37]. The lower reported values for dietary micronutrients may, in part, be attributed to the underreporting of foods consumed less frequently, which may serve as primary sources of certain micronutrients”.
Additionally, it would be useful to know how the dietary intake of micronutrients was adjusted for factors like total calorie intake or food group consumption.
Response: Good point. Given that the objective of this manuscript is to estimate the prevalence of dietary micronutrient intake among participants who do or do not use dietary supplements, diet is not considered in this study as an exposure factor of interest that may be potentially associated with a health/disease outcome. The primary goal of adjusting nutrient and food intake for total dietary energy in epidemiological studies is to assess how variations in diet composition impact health outcomes. In this sense, we follow descriptive methodologies similar to those employed in NHANES and the Household Budget Survey (POF), which are discussed in the manuscript, and present the unadjusted mean values ​​of micronutrient intake estimated from the two 24-hour dietary recalls (24hR).
The study does not suggest public health considerations and which efforts should be made to regulate supplement use, particularly among individuals who already meet their nutritional needs through diet? A discussion on the policy implications could enhance the relevance of the study’s conclusions.
Response: In lines 285-292, we address key topics essential for understanding the dietary habits of the Brazilian population. We have revised the conclusion to more clearly highlight the specific contributions of our findings to Brazilian public health.
The conclusion seems repetitive, reiterating the “inverse supplement hypothesis” without offering new insights or a strong conclusion on how this finding should be interpreted in terms of action or further research. The ending could be more impactful by proposing future directions for research or policy recommendations based on the findings.
Response: Thank you for the careful review. We revised the conclusion according to your suggestion.
Round 2
Reviewer 4 Report
Comments and Suggestions for Authors
The author has made all the revisions to the manuscript in response to my comments.